# The Presence of *Mycobacterium leprae* in Wild Rodents

**DOI:** 10.3390/microorganisms10061114

**Published:** 2022-05-28

**Authors:** Maxwell Furtado de Lima, Maria do Perpétuo Socorro Amador Silvestre, Everaldina Cordeiro dos Santos, Lívia Caricio Martins, Juarez Antônio Simões Quaresma, Bruno de Cássio Veloso de Barros, Marcos Jessé Abrahão Silva, Luana Nepomuceno Gondim Costa Lima

**Affiliations:** 1Bacteriology and Mycology Section, Evandro Chagas Institute (IEC), Ananindeua 67030-000, PA, Brazil; maxwellfurtado@iec.gov.br (M.F.d.L.); socorroamador@iec.gov.br (M.d.P.S.A.S.); everaldinasantos@iec.gov.br (E.C.d.S.); jesseabrahao10@gmail.com (M.J.A.S.); 2Arbovirology and Hemorrhagic Fevers Section, Evandro Chagas Institute (IEC), Ananindeua 67030-000, PA, Brazil; liviamartins@iec.gov.br; 3Tropical Medicine Center, Federal University of Pará (UFPA), Belém 66075-110, PA, Brazil; juarez.quaresma@gmail.com; 4Faculty of Veterinary Medicine, College School of the Amazon (ESAMAZ), Belém 66023-570, PA, Brazil; brunocvb@yahoo.com.br

**Keywords:** ecoepidemiology, leprosy, *Mycobacterium leprae*, animals

## Abstract

Leprosy is a chronic infection caused by *Mycobacterium leprae*. There is a lack of data regarding environmental reservoirs, which may represent a serious public health problem in Brazil, especially in the state of Pará, which occupies the fourth position in incidence of cases in the country. Previous studies report evidence of infection occurring among armadillos, mangabei monkeys, and chimpanzees. In the present study, wild animals were captured and tested for the presence of anti-PGL-1 antibodies and *M. leprae* DNA. Fieldwork was carried out from October to November of 2016 in the cities of Curionópolis and Canaã dos Carajás, southeast of Pará state. Small and medium-sized wild animals were captured using appropriate traps. A total of 15 animals were captured. Sera and viscera fragments were collected and tested by ELISA and PCR methods. The presence of *M. leprae* DNA was confirmed by sequencing of specific gyrase gene in three animals of two different species, including one *Necromys lasiurus* (liver sample) and two *Proechimys roberti* (kidney and liver samples). This unprecedented finding suggests that species other than those previously reported are responsible for maintaining *M. leprae* in nature.

## 1. Introduction

Leprosy is an ancient, long-term infectious disease caused by *Mycobacterium leprae*. It is considered a neglected disease, mainly reported in low-income regions with poor and basic sanitation, causing serious physical and psychological sequelae in infected individuals. Chronicity of cases is often related to late onset of symptoms, representing a serious public health problem in Brazil, the second country in incidence of cases [1]. 

Several factors contributed to leprosy endemicity in Brazil, including acceleration of migratory movements between Brazilian regions and poor comprehension of natural and environmental reservoirs of the bacilli. Indeed, previous experimental studies evaluating animals demonstrated armadillos as potential reservoirs and sources of transmission of the bacilli, as they are part of the food subsistence in some regions [2].

In the Amazon, the areas of primary, secondary, and grassland forests comprise a remarkable diversity of organisms, sharing the same environment. Small and medium-sized animals such as turtles, reptiles, rodents, and especially marsupials are relevant from an ecological point of view, playing a significant role as seed dispersers, pollinators, regulators of populations of vertebrates, smaller invertebrates, and plants, and occupy basal positions in the trophic web, serving as prey for larger vertebrates [3,4,5,6,7,8]. In particular, several species of primates consume other vertebrates in a similar way to these animals; consumption of these vertebrates by nonhuman primate animals has been observed in Africa (about 22 species of cercopithecoid monkeys and other apes) within a human context of consumption of these species [9,10]. 

Wild animals contribute to the maintenance and spread of several infectious agents, including viral, parasite, fungi, or bacteria pathogens; therefore, animals can be hosts of one or more pathogenic agents causing a wide range of symptoms [11]. *M. leprae* has been previously identified naturally occurring in armadillos (*Dasypus novemcinctus*) in Southern Louisiana; chimpanzees (*Pan troglodytes*), sooty mangabey monkeys (*Cercocebus atys)* in Congo, and cynomolgus macaques (*Macaca fascicularis*) in the Philippines, as well as in wild rodents, such as squirrels [12]. 

In this context, the present study pioneered in this analysis aimed to detect of anti-PGL-1 antibodies and *M. leprae* DNA in viscera of wild rodents in the Brazilian Amazon region. The presence of people in forest sections where *M. leprae*-positive wild animals were detected is concerning due to the risk of zoonotic infection as wild rodents may play a major role in the environmental presence of *M. leprae*, as revealed in this analysis. 

## 2. Materials and Methods

### 2.1. Sampling

The present study was approved by the Ethics Committee of the Evandro Chagas Institute/SVS/MS (N° 006/2014) and animal captures had authorization issued by IBAMA (N° 24/2016–21 March 2016). 

Fifteen wild rodents and marsupials of both sexes were captured at the cities of Curionópolis and Canaã dos Carajás, southeastern region of the state of Pará, an endemic region for leprosy, during a fieldwork mission conducted by the Bacteriology and Mycology Section, Evandro Chagas Institute, from 30 October to 20 November, in an equal period of ten days for each city. 

In each area, the capture of animals was performed using two appropriate types of traps, including one closed aluminum trap for small animals measuring 9 × 7.5 × 23.5, Sherman type (2700/night), and a Tomahawk-type trap with galvanized wires measuring 40 × 15 × 15 cm, for medium-sized animals (1600/night) (Figure 1 and Figure 2).

Traps were placed 2 m apart from each other in areas of primary forests and grasslands in the study location. Tomahawk-type trap contained fruit baits and bacon bits, while Sherman-type trap contained peanut butter, grated cheese, and oat flakes. Each trap was checked daily in the morning with baits replaced every two days, except on rainy days, when bait changes were made daily.

The captured animals (Table 1) were placed in cotton bags to minimize stress and transported to field laboratory for evaluation. Initially, captured animals were weighed and sedated with anesthetic (zolethyl), followed by species identified based on morphology features as described in dichotomous keys and illustrative literature [4,13,14]. Biometry and biological samples such as blood, fragments of viscera, lymph node, reproductive organs, and ear tissue were recovered from euthanized animals, including those that did not resist the sedation procedure and with suspected diseases. Non-euthanized animals were released in the same area of capture after full recovery from anesthesia effect. 

Blood samples were collected using disposable syringes and needles, followed by centrifugation for sera obtention; viscera fragments were recovered by using individual scissors and stainless-steel forceps. Samples were individually identified, stored into two mL plastic cryotubes, kept under refrigeration within containers with liquid nitrogen (at −196 °C), and transported to the Leprosy Laboratory, Bacteriology and Mycology section (IEC), where they were stored at −70 °C for further assays. 

### 2.2. ELISA Method

The detection of IgM antibodies against *M. leprae* was performed by means of a standard enzyme-linked immunosorbent assay (ELISA), using phenolic glycolipid I (PGL-1) antigen. It was applied as a semi-synthetic antigen that presents an immunogenic trisaccharide terminal portion of the PGL chain-I, denominated natural trisaccharide phenylpropionyl bovine serum albumin (NT-P-BSA), lyophilized 100 µg/mL. Hence, it was eluted with 100 µL of buffered and sterilized Milli-Q water, and disaccharide antigen (Di), ND-O-BSA. ND-O-BSA (250 µg/mL) was eluted in 250 µL of sterile milli-Q water and combined with 100 µg/mL of Di added (blended) to natural trisaccharide analogue + 100 µL of trisaccharide. Antigens were diluted to concentration of 1:250 (antigen concentration: 0.01 µg of sugar/mL), followed by 1:200 dilution in ammonium hydrogen carbonate (NH_4_)HCO_3_ buffer solution. 96-well U-shaped plates were sensitized with the antigen, which were used to measure the levels of anti-PGL-I IgM antibodies in the sera samples. According to previous standardization, samples reaching an Optical Density (OD) equal or higher to 0.2 were considered positive [15]. 

### 2.3. Molecular Methods

DNA extraction was performed from animals’ viscera fragments using the kit DNeasy Blood & Tissue (Qiagen, Hilden, Germany), following the manufacturer’s guidelines. Amplification of the specific genomic region—*M. leprae repetitive element* RLEP2 (NCBI X17151)—of 238 bp was performed by polymerase chain reaction (PCR) on Veriti thermocycler (Applied Biosystems, Foster City, CA, USA) using three primers and two PCR reactions, as follows: RLEP2.1 (5′ATATCGATGCAGGCGTGAG3′) and RLEP2.2 (5′GGATCATCGATGCACTGTTC3′) for the first PCR, and primers RLEP2.2 (5′GGATCATCGATGCACTGTTC3′) and RLEP2.3 (5′GGGTAGGGGCGTTTAGTGTGT3′) for the second PCR [16]. Amplifications were performed under the following conditions: initial denaturation at 94 °C for 5 min, followed by 40 cycles at 94 °C for 30 s for denaturation, 59.5 °C for 30 s for annealing, 72 °C for 1 min for extension, and final extension at 72 °C for 14 min. For the second reaction annealing temperature at 59.8 °C was applied. Samples were kept at 4 °C until they were removed from the thermocycler and stored at −20 °C. PCR products were subjected to electrophoresis on 2.0% agarose gel in 1x TBE added of 3μL SYBR™ Safe for 30 min at 120 V; Amplicons bands of 282 bp and 238 bp for each reaction, respectively, were visualized under ultraviolet in transluminator equipment. 

Amplification of *gyrA* region (GenBank accession NC002677) was performed using the primers *gyrA*F (5′CCCGGACCGTAGCCACGCTAAGTC3′) and *gyrA*R (5′CATCGCTGCCGGTGGGTCATTA3′), under the following thermocycling conditions: initial denaturation at 94 °C for 5 min; six cycles at 94 °C for 45 s, 68 °C to 63 °C for 45 s, and 72 °C for 90 s; followed by 35 cycles at 94 °C for 45 s, 62 °C for 45 s, and 72 °C for 90 s, and final extension at 72 °C for 10 min. PCR products of 187 bp were purified using the EasyPure PCR Purification Kit (Transgen, Beijing, China), according to the manufacturer’s recommendations. Reaction products were bidirectionally sequenced using Big Dye Terminator v3.1 chemistry. Each reaction was composed of 10.5 μL of nuclease-free water, 2 μL of buffer, 0.75 μL of Big Dye, 0.75 μL of primer (*gyrA* at 5 pmols), and 2 μL of purified PCR product, and submitted to amplification under the following conditions: initial denaturation at 96 °C for 1 min, followed by 25 cycles at 96 °C for 10 s, annealing at 50 °C for 5 s, and extension at 60 °C for 4 min. The resulting product were purified using Big Dye X-terminator kit and sequenced on ABI 3130 Genetic Analyzer sequencer (Applied Biosystems, Foster City, CA, USA). Obtained sequences were compared to those available at BLAST platform (National Center for Biotechnology Information—NCBI, Bethesda, MD, USA) website. Sequencing of *gyrA* gene was performed to confirm RLEP2 fragment PCR positive results. The *gyrA* sequence (GenBank accession NC002677) was used as reference. Obtained sequences were not deposited in GenBank database, since in GenBank complete genomic sequences are inserted.

## 3. Results

A total of six (6) and nine (9) animals were captured in the cities of Curionópolis and Canaã dos Carajás, respectively, from which 15 serum samples and 30 viscera samples were obtained (Table 1). The animals were identified using morphological features in the field, followed by confirmation by means of mitochondrial DNA sequencing in the laboratory. Rodent species were the most frequent in the present study.

ELISA results for the 15 sera tested were negative for the anti-PGL-1 antibody, although some samples showed results close to the 0.2 *cutoff*. Among the 30 viscera samples, RLEP2 *M. leprae* region was amplified from samples of three different animals, two being exclusively wild species: (I) liver sample from *Necromys lasiurus* species, (II) kidney sample *Proechimys roberti* species, and (III) liver sample from *Proechimys roberti* species (Figure 3, Figure 4, Figure 5 and Figure 6).

Only three (3) samples were positive in PCR and were submitted to Sanger sequencing from the *gyrA* region, which confirmed the presence of *M. leprae* DNA. Hence, all negative samples in PCR were amplified after the addition of 1 picogram (pg) of *M. leprae* DNA. 

## 4. Discussion

Anthropogenic impacts on natural environment cause destruction and fragmentation of natural habitats, forcing wild animals to seek smaller, discontinuous, and vulnerable refuges in surrounding areas of urban environments [17]. Human expansion in tropical forests is a major health risk factor, especially due to the emergence or re-emerging zoonoses [18].

Although *M. leprae* is considered to be primarily hosted by humans, wild animals infected with *M. leprae* have been considered for more than three decades, and the investigation and detection of nonhuman reservoirs has intended to contribute with the comprehension of leprosy transmission patterns and control strategies in endemic regions [19,20].

The areas of capture, which are located in the Amazon region, are considered highly anthropized, with changes in the environmental landscape due to deforestation and timber extraction, resulting in a reduction in plant biodiversity, forest fragmentation, and ecological disturbance. In this context, the interaction of wild animals with domestic animals and humans is intensified and may potentially cause the occurrence of zoonotic outbreaks [21,22,23].

The use of molecular methods aiming specific targets has contributed to the present investigation, demonstrating the presence of *M. leprae* bacilli in the environment and possible reservoirs and/or sources of infection [24,25,26,27]. This study applied two highly specific markers of *M. leprae*, including RLEP2 for DNA detection by PCR and *gyrA* for confirmation by Sanger sequencing of RLEP2-positive samples.

Data obtained in the present study show nonprimate wildlife species as reservoirs of *M. leprae* in the Brazilian Amazon region. Previous reports describe nonprimate wild animals hosting *M. leprae*, including armadillos belonging to the *Dasypus novemcinctus* species, first reported by [28] in Louisiana (U.S) in the year 1977. In 2009, *M. leprae* DNA was detected on sera, ear tissue, and spleen samples from PGL I-positive armadillos in Colombia, followed by reports in the Southeast (2010) and Northeast (2012) regions of Brazil, suggesting disease prevalence rates over 20% in animals from locations other than the U.S [29,30,31,32]. 

All 15 animals analyzed in the present study were captured in nearby forest areas with human activity. Reptiles, marsupials, and mammalian rodents were analyzed; however, only among rodent species was *M. leprae* DNA detected. Rodent species represented the majority of animals captured in the present study, which may be related to the high adaptability capacity of such species to anthropogenic impacts caused by humans [33,34,35]. Rodents are animals found in forest fragments, fields, gallery forests, and urban perimeters, as they occupy several ecological niches [33,34,35,36,37]. Furthermore, previous data highlight their major public health relevance, as they tend to harbor and serve as vectors of zoonotic pathogens, causing diseases such as leptospirosis, hantavirosis, and leishmaniasis [38,39,40,41,42,43,44]. Finally, given the synanthropy of rodent species such as *Rattus rattus*, these animals are often associated with diseases caused by mycobacteria species [45,46,47].

Positive samples were detected from liver and kidney organs, suggesting that *M. leprae* follows an infection pattern similar to that reported in armadillos (*Dasypus novemcinctus* and *Euphractus sexcinctus*), as well as being the first report of *M. leprae* DNA in such organs [30,31,32]. 

A study evaluating squirrels belonging to the *Ictidomys tridecemlicleanus* species revealed the presence of acid-fast bacilli, indicating multiplication and susceptibility to *M. leprae* infection [48]. Another report with *Sciurus vulgaris* species in England and Scotland described the presence of a dermatitis in six squirrels, as well as bilateral areas of alopecia and cutaneous edema in the muzzle, lips, eyelids, ear pinnae, and distal parts of limb areas. Histopathology methods applied in samples from three squirrels demonstrated the presence of granulomatous dermatitis, epithelioid macrophages, globias, and acid-fast bacilli; PCRs from skin dermatitis samples were also positive for *M. leprae* DNA [49]. Also in England, *M. leprae* DNA was detected in 21 wild squirrels without presentation of clinical signs of leprosy [50]. To date, no evidence for *M. leprae* infection in squirrels or wild rodents outside the United Kingdom has been reported. A study in Mexico, a location geographically closer to Brazil, was conducted evaluating *Neotoma albigula* species; however, all 72 samples analyzed were negative [51]. Therefore, the present study is the first to report *M. leprae* DNA in wild rodents from South America, as well as outside the United Kingdom. Even though such findings are not definitive evidence that rodents act as a source of infection for humans, the present data support the hypothesis of zoonotic transmission suggested by several authors [30,31,32,48,50,51]. 

There is still a lack of data regarding the incubation period and symptoms presentation in wild animals; however, infection by *M. leprae* presented an onset of symptoms after 15 months in experimental settings [45,52]. In addition, it is rare to find wild animals that are visibly multibacillary, probably due to their poor health progression, short lifespan, and prey activity [53]. RLEP2-positive animals may have a significant role in active and recent dispersal dynamics, as compared to humans, symptoms may appear as early as two to seven years after infection [54].

The origin of *M. leprae* infection is still not clear in the present study. Are such bacilli ubiquitously distributed in the Amazon region? Or does the human presence, as well as individuals with leprosy nearby environmental settings, favor the bacilli spread?

Today, with more robust methodologies, it is possible to address questions regarding the relationship of the surrounding environment and spread of *M. leprae*, contributing with knowledge on its transmission. The present study suggests that animals other than the armadillos and primates may also be responsible for the maintenance of *M. leprae* in nature settings.

## 5. Conclusions

The presence of *M. leprae*-infected animals in forest remnants suggests the existence of active and recent bacillary transmission, even considering the short longevity of such animals and slow pathogen multiplication. The frequent presence of humans in forest fragments where *M. leprae*-positive wild animals were found may contribute to zoonotic infections. Furthermore, areas under human intervention in nature, mainly in those with mineral exploration activities in the region, need to be investigated. The PCR methodology aiming the detection RLEP2 gene, along with confirmation by *gyrA* sequencing, was demonstrated as an effective way to detect the *M. leprae* DNA in wild rodents and other wild life species.

## Figures and Tables

**Figure 1 microorganisms-10-01114-f001:**
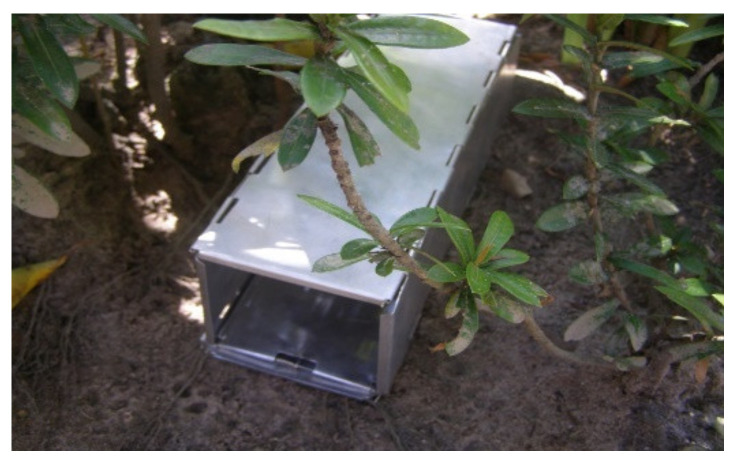
Sherman-type trap.

**Figure 2 microorganisms-10-01114-f002:**
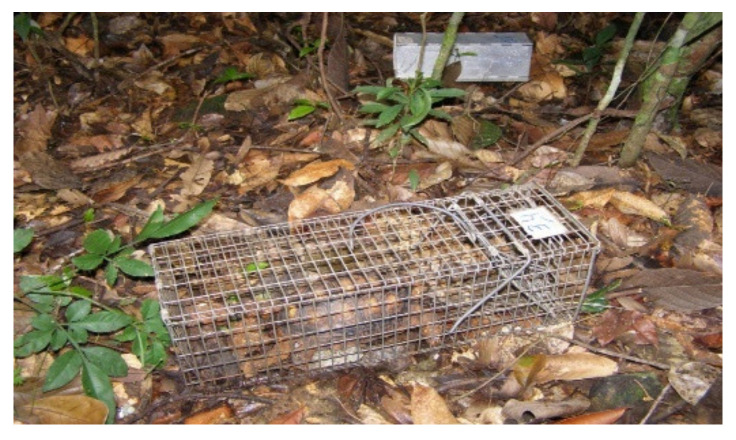
Tomahawk-type trap.

**Figure 3 microorganisms-10-01114-f003:**
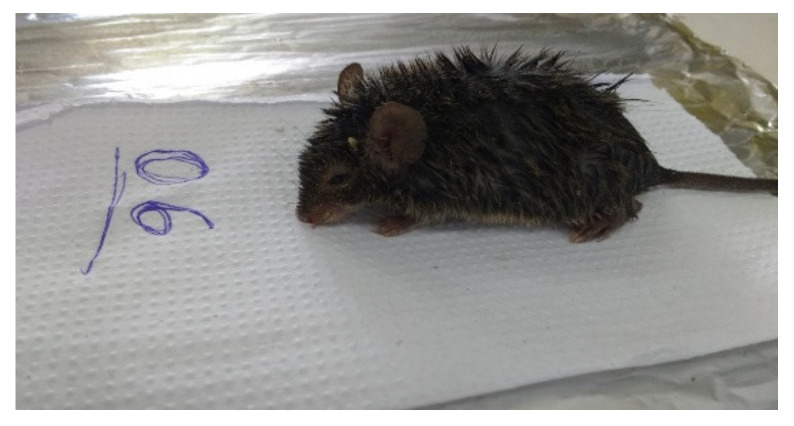
Necromys lasiurus.

**Figure 4 microorganisms-10-01114-f004:**
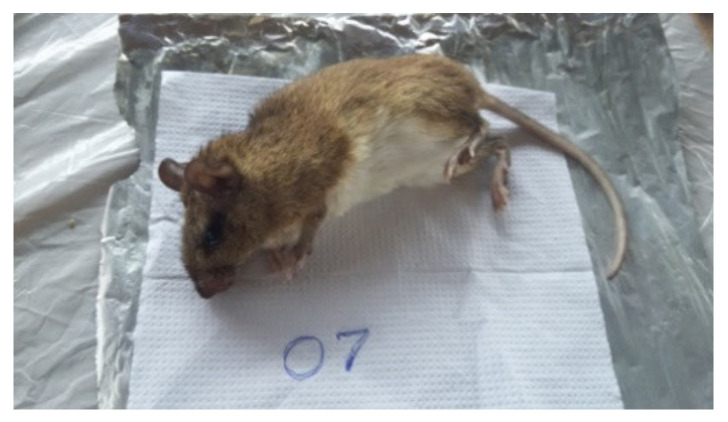
Proechimys roberti.

**Figure 5 microorganisms-10-01114-f005:**
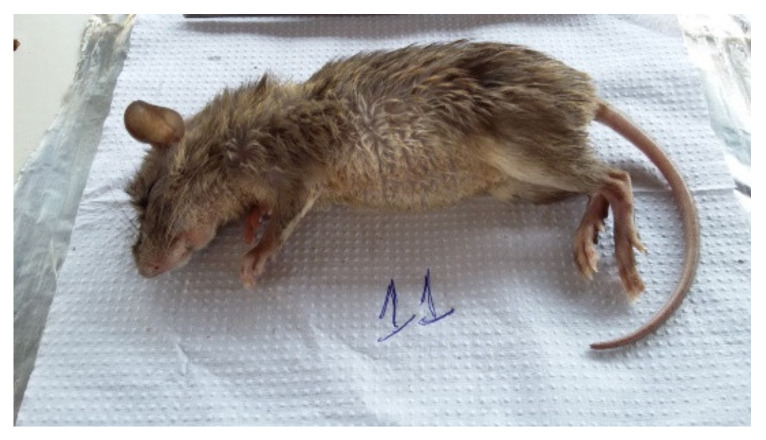
Proechimys roberti.

**Figure 6 microorganisms-10-01114-f006:**
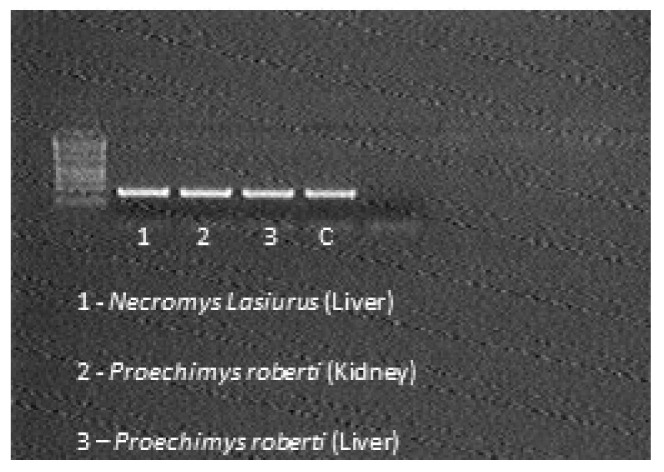
Positive PCR.

**Table 1 microorganisms-10-01114-t001:** Animals’ information according to capture location.

N°	Municipality	Date	Specie	PCR	Trap/Number and Line
1	Curionópolis/Pará—Area Serra Leste—Location: Farm (Water Collection)	2 November 2016	*Necromys lasiurus*	Neg	Sherman—147/Line 4
2	Curionópolis/Pará—Area Serra Leste—Location: Farm (Water Collection)	3 November 2016	*Necromys lasiurus*	Neg	Sherman—141/Line 4
3	Curionópolis/Pará—Area Serra Leste—Location: Farm (Water Collection)	3 November 2016	*Ameiva ameiva*	Neg	Sherman—130/Line 4
4	Curionópolis/Pará—Area Serra Leste—Location: Farm (Water Collection)	4 November 2016	*Ameiva ameiva*	Neg	Sherman—36/Line 3
5	Curionópolis/Pará—Area Serra Leste—Location: Farm (Water Collection)	5 November 2016	*Necromys lasiurus*	Neg	Sherman—141/Line 4
6	Curionópolis/Pará—Area Serra Leste—Location: Farm (Water Collection)	9 November 2016	*Necromys lasiurus*	Pos	Sherman—143/Line 4
7	Canaã dos Carajás/Pará—Area Serra Sul—Location: Plateau S11D	11 November 2016	*Proechimys roberti*	Pos	Tomahawk—16/Line 1
8	Canaã dos Carajás/Pará—Area Serra Sul—Location: Plateau S11D	12 November 2016	*Marmosops parvidens* (Br-1311)	Neg	Sherman—04/Line 1
9	Canaã dos Carajás/Pará—Area Serra Sul—Location: Plateau S11D	12 November 2016	*Proechimys cf. roberti*	Neg	Sherman—16/Line 1
10	Canaã dos Carajás/Pará—Area Serra Sul—Location: Plateau S11D	12 November 2016	*Oxymycterus amazonicus*	Neg	Sherman—78/Line 1
11	Canaã dos Carajás/Pará—Area Serra Sul—Location: Plateau S11D	12 November 2016	*Proechimys roberti*	Pos	Sherman—131/Line 2
12	Canaã dos Carajás/Pará—Area Serra Sul—Location: Plateau S11D	15 November 2016	*Marmosops parvidens* (Br-1313)	Neg	Sherman—17/Line 1
13	Canaã dos Carajás/Pará—Area Serra Sul—Location: Plateau S11D	15 November 2016	*Didelphis aurita* (1314)	Neg	Tomahawk—34/Line 1
14	Canaã dos Carajás/Pará—Area Serra Sul—Location: Plateau S11D	15 November 2016	*Proechimys roberti*	Neg	Tomahawk—64/Line 1
15	Canaã dos Carajás/Pará—Area Serra Sul—Location: Plateau S11D	17 November 2016	*Marmosops parvidens* (Br-1315)	Neg	Sherman—12/Line 1
**Total**	**15**				

## Data Availability

The data presented in this study are available in inserted article.

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
