# Peer review of "The Presence of Mycobacterium leprae in Wild Rodents"

_microorganisms, 2022, doi:10.3390/microorganisms10061114_

Round 1

Reviewer 1 Report

The Presence of Mycobacterium leprae in Wild Rodents

The authors investigated the occurrence of Mycobacterium leprae in wild rodents in Brazil. They captured 15 individuals at two locations and found the infection in three individuals. Generally, the paper presents interesting and important novel results, however the biggest weakness of this work is the quality of writing. There are numerous grammatically incorrect phrasing and confusing sentences throughout the text. I strongly suggest revising it. Otherwise, I only have some minor comments, which the authors can find below.

P1, L19-20: please add the scientific name of these species and also add locality (e.g. Africa).

P1, L20-21: this sentence is confusing and grammatically incorrect, please re-phrase.

P1, L29: species name in italics

P3, Tables: I don’t understand why these two tables are separated. Seems to me it shows the same type of data, therefore I suggest combining them.

P6, L226: italicize the species name, please (and also throughout the text)

Author Response

To Reviewer 1:

Point 1: P1, L19-20: It was corrected.

Point 2: P1, L20-21: The changes were made.

Point 3: P1, L29: The modifications were done.

Point 4: P3, Tables: The Tables have been unified.

Point 5: P6, L226: The corrections were made.

Reviewer 2 Report

29 line: "M. leprae" must be written Italic. 

46-47 lines: turtles and reptiles are not mammals.

33-59 lines: the introduction pays a lot of attention to the monkeys and there is nothing about the rodents that are the object of study. The purpose of the work is not defined. Also the main conclusions are not presented.

74-77 lines: Figures 1 and 2 are not informative. You can remove them.

92-93 lines: Why did you separate the Tables 1 and 2? Move the data to a single table.

166 line: "M. leprae" must be written Italic. 

171-176 lines: the titles of the pictures 3-5 are from the Latin name of the species. Its must be written Italic. 

179-181 lines: how many samples were sequenced? Are the sequences submitted to the GenBank database? If so, provide the accession numbers. It would also be helpful to include a phylogenetic tree constructed on the basis of the sequences in the results.

197 line: "M. leprae" must be written Italic. 

211 line: "All 21 animals analyzed in this study...". Only 15 animals are mentioned in the methods and results. 

222 line: "Rattus rattus" must be written Italic. 

226 line: "Proechimys roberti" must be written Italic. 

232 line: "M. leprae" must be written Italic. 

Author Response

To Reviewer 2:

Point 1: 29 line: It was corrected.

Point 2: 46-47 lines: The information was corrected.

Point 3: 33-59 lines: Alterations were done.

Point 4: 74-77 lines: We decided to maintain these figures for the best understanding of the reader.

Point 5: 92-93 lines: It was corrected.

Point 6: 166 line: Corrected as requested.

Point 7: 171-176 lines: Modifications have been done.

Point 8: 179-181 lines: The information was added in the text. “Only positive PCR samples were sequenced. From these samples, the Sanger sequencing of only one gyrA region was performed. We performed this sequencing to confirm the positive result of the PCR that amplified the RLEP fragment. In this way, we do not submit to the GenBank database, since in GenBank complete genomic sequences are inserted.”

Point 9: 197 line: Alteration was done.

Point 10: 211 line: It was corrected.

Point 11: 222 line: Corrections were made.

Point 12: 226 line: Corrected as requested.

Point 13: 232 line: The modifications were done.